# Effect of Catadioptric Component Postposition on Lens Focal Length and Imaging Surface in a Mirror Binocular System

**DOI:** 10.3390/s19235309

**Published:** 2019-12-02

**Authors:** Fuqiang Zhou, Yuanze Chen, Mingxuan Zhou, Xiaosong Li

**Affiliations:** Key Laboratory of Precision Opto-Mechatronics Technology, Ministry of Education, Beihang University, Beijing 100083, China; chenyuanze@buaa.edu.cn (Y.C.); zhoumx@buaa.edu.cn (M.Z.); lixiaosong@buaa.edu.cn (X.L.)

**Keywords:** cameras, binocular vision, catadioptric stereovision, optical path analysis

## Abstract

The binocular vision system is widely used in three-dimensional measurement, drone navigation, and many other fields. However, due to the high cost, large volume, and inconvenient operation of the two-camera system, it is difficult to meet the weight and load requirements of the UAV system. Therefore, the study of mirror binocular with single camera was carried out. Existing mirror binocular systems place the catadioptric components in front of the lens, which makes the volume of measurement system still large. In this paper, a catadioptric postposition system is designed, which places the prism behind the lens to achieve mirror binocular imaging. The influence of the post prism on the focal length and imaging surface of the optical system is analyzed. The feasibility of post-mirror binocular imaging are verified by experiments, and it is reasonable to compensate the focal length change by changing the back focal plane position. This research laid the foundation for the subsequent research on the 3D reconstruction of the novel mirror binocular system.

## 1. Introduction

With the rapid development of computer vision and imaging process technology, binocular vision system has been widely applied to different fields, such as industrial manufacturing, parts and workpiece inspection [1,2,3], drone navigation [4,5], intelligent security, and so on. Additionally, the demand of three-dimensional measuring technology has been continuously increasing these years, so the research on stereo camera system that can obtain the three-dimensional surface of objects has attracted great attention [6]. For example, Rodríguez and Mejía Alanís [7] discussed the situation that the first camera captures the occluded regions of the second camera, and the second camera captures the occluded regions of the first camera in a binocular system. Traditional binocular stereo vision system is generally composed of two cameras to capture scenes of the object from different positions [8], which is costly, time consuming, and of poor synchronization [9]. The delay in capturing images makes it impossible for both cameras to take pictures at the same time. Besides, the weight and data load of the two-camera measurement system are large, which is difficult to meet the volume and weight requirements of flight measurement in UAV navigation. To solve the problems above, many researchers have studied the mirror binocular system consisting of the single camera and catadioptric components.

Compared with the traditional stereo vision composed of multiple cameras, the mirror binocular system has great superiority. This method use the single camera and catadioptric components to acquire stereo vision images. This kind of vision sensors is low-cost, simple to operate, perfectly synchronous, and fast for measurement. Gluckman and Nayar [10] first proposed a catadioptric stereo system using a plane mirror. By using specular reflection to capture a stereo image in a measurement scene, the data acquisition and calibration process of the vision sensor is simplified. Feng X F and Pan D F et al. [11] proposed a stereo vision sensor model based on plane mirror imaging, and discussed the measurement theory of 3D points, which is suitable for high precision measurement in short distance. Yu L and Pan B et al. [12] designed a stereo camera system with a single camera combined with a four-plane mirror to capture a pair of object images with fast speed and high robustness. Zhou F et al. [13] designed a mirror binocular system consisting of a single camera and a quadrangular pyramid-shaped plane mirror group to establish a single camera multi-mirror polar constraint model. DooHyun Lee and InSo Kweon et al. [14] proposed a new stereo camera system using double prisms, in which a triangular prism is placed in front of the lens to construct a simple stereo system. It greatly simplifies the calibration process of the system. Xiao Y and Lim K B et al. [15] designed a single-lens stereo vision system based on multiface prisms, and extended from trinocular to multiocular, and established two models to describe the vision system. Chen C Y and Deng Q L et al. [16] proposed a single-lens panoramic stereo photography using a double-symmetric prism placed in front of the lens to obtain images of four different viewing angles, which makes the image resolution enhanced several times and eliminates the moiré effect. Teresa Wu et al. [17] explored the use of liquid-crystal shutter-based technology for stereoscopic display of prism-based single-camera systems for stereoscopic display, quantitatively assessing the performance of the proposed system compared with a benchmark dual camera system. Chen C Y and Yang T T et al. [18] designed a microprism array optical system to realize the acquisition of single camera stereo image pairs, and explained the principle of the optical system for capturing stereo images. Also, the field of view is optimized and the lens is optimized. Zhou F and Chen X et al. [19] proposed an asymmetric binocular catadioptric system, analyzed the influence of optical-path-difference in the optical system, and compensated it.

Ray tracing is an analytical method for accurately calculating the exact path of light passing through an optical system by applying the law of refraction to each refraction surface when actually processing optical system imaging problems or optical design. Yu L and Wei Q et al. [20] presented a compact structure composed of two transmittable aspheric surfaces and two reflective conicoid surfaces based on optical design, which enhanced the stability of the second mirror and has good imaging ability. Hagimori H et al. [21] explained the changes in the optical design concept of the focusing function in the zoom lens, and introduced the development of the zoom lens and the different types of typical lenses. Shen T C and Drost R J et al. [22] proposed a catadioptric beacon position system that can provide mobile network nodes with omnidirectional situational awareness of neighboring nodes. Gagnon Y L and Speiser D I et al. [23] proposed an optical system characterization method that simplifies numerical ray tracing, which greatly simplified the calculations for the location and direction of the rays by using the Chebyshev approximation, whereas all of the above existing mirror binocular researches place the catadioptric structure in front of the lens, making the size of the measuring sensor system larger, and the convenience of measurement has not been significantly improved.

This paper proposed a novel single-camera stereo vision system with a catadioptric component placed behind the lens based on the existing mirror binocular, and we concentrate on the influence of the postposition of catadioptric component on the focal length and imaging surface in whole system, thus making the single-camera stereo vision system capable of imaging by focal length compensation. In this paper, we use a small wedge angle prism as a catadioptric component and place it between the lens and camera, which greatly reduces the size of the measurement system and the cost with perfect synchronization. However, at the same time, it will change the focal length of the optical system and the position of the imaging surface, so that the imaging surface cannot fall on the original CCD plane, resulting in defocusing and poor image quality. Therefore, the change of the focal length and the position change of the imaging surface are studied to compensate, so that the image surface, which has been separated into left and right images, can fall on the CCD surface and acquire a clear image. Thus, we can verify the feasibility of single-camera binocular vision system. To verify the feasibility of the sensor, we carried out the calibration experiment. This article concentrates on the effect on the focal length and imaging surface, and we aim to enable the sensor have the ability to image. The three-dimensional reconstruction contents are planned to be discussed in future work.

The rest of the paper is organized as follows. Section 2 provides the detailed descriptions of the proposed system and the derivation of ray trace. Section 3 is followed by the experiments and analysis with setup parameters. Finally, Section 4 concludes the paper and proposes further research areas.

## 2. Principle

### 2.1. Influence of Catadioptric System Postposition on the Incident Light Path of Parallel Light

The camera used in machine is composed of a lens and a CCD camera. The camera imaging principle is similar to pinhole imaging model, in which the lens is equivalent to the pinhole [24]. It can receive enough light to make the camera get proper exposure in a short time, and it is capable of concentrating the beam, imaging a space object onto camera CCD plane to obtain a clear image, which can be seen by the Gaussian imaging formula. We can simplify the complex structure of the lens into a single convex lens for analysis. In the past, the design concept of the mirror binocular system was to place the catadioptric structure in front of the lens, as shown in Figure 1.

In Figure 1a, a triangle prism is placed in front of the camera lens. The prism surface will produce a refraction phenomenon, thus changing the optical path to form two beams to achieve the effect of the image. The image acquisition result can be equivalent to the effect of shooting by the left and right virtual cameras in the figure. Figure 1b–d show that the plane mirror is placed in front of the lens, and the light path is changed by the reflection of the plane mirror to form a beam splitting effect. Figure 1b shows a tilting single plane mirror placed in front of the lens, which is equivalent to a binocular system consisting of a real camera and a virtual camera. Figure 1c uses a four-plane mirror to form a catadioptric structure, which the light beam is split by two reflections. Figure 1d uses a quadrangular pyramid plane mirror to form a catadioptric structure. Both Figure 1c,d are equivalent to a binocular system consisting of two virtual cameras.

In the case above, the optical path is changed outside the camera system. Therefore, it will not affect the focus distance, and the position of the imaging surface in the camera does not change, either. We usually use virtual cameras to simulate for analysis.

In this paper, the catadioptric system uses the prism shown in Figure 1a and places it between the lens and the CCD plane of the camera to form a catadioptric postposition optical system, as shown in Figure 2. Light splitting phenomenon occurs inside the camera. As a result, the optical path in the original camera pinhole imaging model is changed, and the camera focal length is also changed. We have carried out simulation analysis and formula derivation.

Figure 3a shows that we use Zemax to simulate the imaging optical path in which a lens is replaced by a single convex lens. We simulate the optical path that parallel lights incident on a single lens, and converge on the focus point. The light lines hit the imaging plane. To observe what kind of change the optical path of the camera optical system will be caused as a result of prism postposition, we create a prism to simulate in Zemax. The position of prism and lens and the change of optical path is shown in Figure 3b. The parameters of each surface of the new lens prism group are set as shown in Table 1.

To quantitatively analyzed the change of focus, we calculate the optical path in the Figure 3b and derive formula. The schematic diagram of the optical path derivation is shown in Figure 4.

The single lens aperture is *D*${}_{1}$, the prism aperture is *D*${}_{2}$, and the distance between lens and prism is *d*. The two beams of the same side parallel light are traced. The distance between the beam 1 and the optical axis is *l*${}_{1}$, and the distance between the beam 2 and the optical axis is *l*${}_{2}$. The local length is *f*, and the wedge angle of the triangular prism is θ. J and K are the exit points of the two rays passing through the single lens. B and D are the incident points of the two rays at the prism. C and G are the exit points of the rays off the prism. α${}_{1}$ and α${}_{2}$ are the incident angles of the first refraction when two rays are incident on the lens. β${}_{1}$ and β${}_{2}$ are the exit angles of the first refraction. *i*${}_{1}$ and *i*${}_{2}$ are the incident angles of the second refraction. *r*${}_{1}$ and *r*${}_{2}$ are the exit angles of the second refraction. F’ is the intersection of the two rays refracted by the prism. The refractive index of the prism material is *n*. Establish a coordinate system at the center of the prism in Figure 4.

Analyze ray 1, it is known by the geometric relationship shown in the figure that
(1)$$\alpha_{1}=\arctan\frac{l_{1}}{f}$$The first refraction occurs when light rays enter the prism, as can be seen from the Snell’s law of refraction:(2)$$\sin\alpha_{1}=n\sin\beta_{1}$$

The coordinate of point *B* is (*d*, *l*${}_{1}$ − *d*tanα${}_{1}$). The slope of line *l*${}_{BC}$ is −tanβ${}_{1}$. Then the equation of the line *l*${}_{BC}$ is *y* − (*l*${}_{1}$ − *d*tan**α**${}_{1}$) = −tanβ${}_{1}$(*x* − *d*). The coordinate of point A is (*d*, $\frac{D_{2}}{2}$). The slope of line *l*${}_{BC}$ is −cotθ. Then, the equation of the line *l*${}_{AP}$ is *y*−$\frac{D_{2}}{2}$ = −cotθ(*x − d*). The simultaneous linear equation can solve the *C* point coordinates as
(3)$$\left\{\begin{array}{c} x_{C}=\frac{D_{2}/2-l_{1}+d\tan\alpha_{1}}{\cot\theta -\tan\beta }+d\\ y_{C}=\frac{D_{2}}{2}-\cot\theta \frac{D_{2}/2-l_{1}+d\tan\alpha_{1}}{\cot\theta -\tan\beta } \end{array}\right.$$
Known by the geometric relationship:(4)$$i_{1}=\frac{\pi }{2}-\left(\pi -\beta_{1}-\left(\frac{\pi }{2}+\theta \right)\right)=\beta_{1}+\theta$$
The second refraction occurs when the light rays exit the prism. Known by the law of refraction: *n*sin*i*${}_{1}$ = sin*r*${}_{1}$. The coordinate of point C is (*x*${}_{C}$,*y*${}_{C}$). The equation of line *l*${}_{CF^{\prime }}$ is *y* − *y*${}_{C}$ = −tan(*r*${}_{1}$ − θ)(*x* − *x*${}_{C}$). The coordinate of point G is (*x*${}_{G}$, *y*${}_{G}$). The equation of line *l*${}_{G}F^{\prime }$ is *y* − *y*${}_{G}$ = −tan(*r*${}_{1}$−θ)(*x* − *x*${}_{G}$). Combining the equation of line *l*${}_{CF^{\prime }}$ and line *l*${}_{GF^{\prime }}$, the coordinates of the intersection of the two rays are solved as follows.
(5)$$\left\{\begin{array}{c} x_{F}^{\prime }=\frac{y_{G}-y_{C}+x_{G}\tan\left(r_{2}-\theta \right)-x_{C}\tan\left(r_{1}-\theta \right)}{\tan\left(r_{2}-\theta \right)-\tan\left(r_{1}-\theta \right)}\\ y_{F}^{\prime }=y_{C}-\tan\left(r_{1}-\theta \right)\left(x_{F}^{\prime }-x_{C}\right) \end{array}\right.$$
In the simulation, to facilitate the observation and analysis, the prism and lens parameters can be assumed. If *D*${}_{1}$ = 25 mm, *D*${}_{2}$ = 25 mm, *f* = 100 mm, *n* = 1.575, *d* = 10 mm, *l*${}_{1}$ = 10 mm, *l*${}_{2}$ = 6 mm, and θ = $10^{{}^{\circ}}$, the calculation result is as follows. The coordinates of point C is (10.6244, 8.9589), and the coordinate of point G is (11.2607, 5.3502). The coordinate of the new focus F’ is (93.2534, −7.3390). The light ray path diagram is shown in Figure 5. The dotted line is the original light path, and the solid line indicates the light path after the prism is added.

To study the effects of changes in prism and optical system parameters on the changes of the optical path and system focal length, we change the parameters and observe the optical path and focus changes; the focus coordinate changes are shown in Table 2.

Rows 1, 2, and 3 of the table show the change in the wedge angle of the prism, and the resulting optical path changes as shown in the Figure 6. The larger the wedge angle of the prism, the greater the degree of deflection of the optical path, the larger the coordinate change of the new focus F’, and the smaller the focal length, and the more obvious the imaging effect. Rows 4 and 5 of the table show the change in the distance between the prism and the lens, and the resulting optical path changes as shown in the Figure 7. The smaller the distance, the larger the coordinate change of the new focus F’, and the smaller the focal length, the more obvious the imaging effect. The first and fourth rows show the lens aperture change, and it is understood that the change in the lens aperture does not affect the new focus F’ when the focal length of the single lens does not change.

It can be seen from the above analysis that the prism postposition system will cause the focus to change, and the image separation effect will be formed, and two symmetrical focal points will be formed, and the focal length will become smaller, thus causing the imaging surface to become close, and the defocusing phenomenon occurs before failing on the CCD plane.

### 2.2. Analysis of Aliasing of Two Images after Imaging Separation

Based on the above formula, we continue to analysis the imaging on the CCD image surface that the light of finite objects path through the lens and the catadioptric postposition system. The formula derived above is to calculate the positional change of the focus in the case where parallel is incident. The object is now placed at a finite distance and the positional of the imaged point is calculated.

In Zemax’s optical path design, we change the ∞ in the object plane (OBJ) thickness column to a finite value, and change the field of view height in Field Data, thus changing the angle of the incident ray, so that we can simulate the optical path of the object through the optical system at a finite distance. The parameters of each surface are shown in Table 3. Layout 3D diagram are shown in Figure 8.

It can be seen from the Figure 8 that the red line and purple lines, respectively, represent the light emitted by the upper and lower ends of the object, and after the system, there are two inverted images on the CCD image surface. To make it a complete and non-aliased image, it is necessary to satisfy that the rays 1 and 4 do not exceed the CCD image plane, and the position of the ray 2 is higher than ray 3 in the image plane. As this optical path is symmetrical, only one side of the optical path need to be analyzed.

In Figure 9, the ray 1 emitted from the top of the object is incident on the CCD plane through the lens and the prism at point D. The *y*-direction coordinate is *y*${}_{1}$, and the ray 2 is incident at the M point, whose *y*-direction coordinate is $y_{2}$. If aliasing does not occur, *y*${}_{1}$ > 0 must be satisfied. To make the image within the CCD plane range, *y*${}_{2}$ > −*D*${}_{CCD}$/2 is required. The *D*${}_{CCD}$ is the *y*-direction dimension length of the CCD plane.

In Figure 9, we can see by Gaussian formula:(6)$$\frac{1}{l^{\prime }}-\frac{1}{l}=\frac{1}{f^{\prime }}$$
Vertical axis magnification:(7)$$\beta =\frac{l^{\prime }}{l}=\frac{h^{\prime }}{h}$$
For ray 1, the angle of incident at point B is α${}_{1}$ = arctan($\frac{\frac{D_{1}}{2}+h^{\prime }}{l^{\prime }}$). It is known by the law of refraction: sinα${}_{1}$ = *n*sin*β*${}_{1}$. The liner equation of BC is *y* − (−$\frac{D_{1}}{2}$ − *d*tanα${}_{1}$) = tanβ${}_{1}\left(x-d\right)$, combining the linear equation of the hypotenuse of the lower half of the prism *y* − (−$\frac{D_{2}}{2}$) = cotθ(x−d). Then, we obtain the coordinates of point C:(8)$$\left\{\begin{array}{c} x_{C}=\frac{D_{1}/2-D_{2}/2-d\tan\alpha_{1}}{\tan\beta_{1}-\cot\theta }+d\\ y_{C}=-\frac{D_{2}}{2}-\cot\theta \frac{D_{1}/2-D_{2}/2-d\tan\alpha_{1}}{\tan\beta_{1}-\cot\theta } \end{array}\right.$$
The incident angle at point C is *i*${}_{1}$ = β${}_{1}$ + θ. Known by law refraction *n*sin*i*${}_{1}$ = *sin**r*${}_{1}$, the linear equation of line CD is *y* − *y*${}_{C}$ = tan(*r*${}_{1}$−θ)(*x*−*x*${}_{C}$); this line intersects the CCD surface at point D. Substituting the abscissa of the CCD surface *x*${}_{D}$ = *D*${}_{CCD}$, the position of the ray 1 on the CCD surface can be obtained *y*${}_{1}$ = *y*${}_{C}$ + tan(*r*${}_{1}$−θ)(*D*${}_{CCD}$−*x*${}_{C}$). Let *y*${}_{1}$ > 0, so that we can make the image not overlap.

For ray 2, the angle of incident at point F is α${}_{2}$ = arctan($\frac{\frac{D_{1}}{2}-h^{\prime }}{l^{\prime }}$). Combining the law of refraction sinα${}_{2}$ = *n*sin*β*${}_{2}$, the coordinates of the point G can be obtained similarly:(9)$$\left\{\begin{array}{c} x_{G}=\frac{D_{2}/2-D_{1}/2+d\tan\alpha_{2}}{\cot\theta -\tan\beta_{2}}+d\\ y_{G}=\frac{D_{2}}{2}-\cot\theta \frac{D_{2}/2-D_{1}/2+d\tan\alpha_{2}}{\cot\theta -\tan\beta_{2}} \end{array}\right.$$

The incident angle at G is *i*${}_{2}$ = β${}_{2}$ + θ. Known by law refraction *n*sin*i*${}_{2}$ = *sin**r*${}_{2}$, the linear equation of line GM is *y* − *y*${}_{G}$ = tan(*r*${}_{2}$−θ)(*x* − *x*${}_{G}$); this line intersects the CCD surface at point M. Substituting the abscissa of the CCD surface *x*${}_{G}$ = *D*${}_{CCD}$, the position of the ray 2 on the CCD surface can be obtained *y*${}_{2}$ = *y*${}_{G}$ + tan(*r*${}_{2}$ − θ)(*D*${}_{CCD}$−*x*${}_{G}$). Let *y*${}_{2}$ > −$\frac{D_{CCD}}{2}$, so that we can make the image within the CCD image plane range.

Based on the above analysis, we design and select the prism parameters. The wedge angle of the prism should not be too large. If it is too large, the total internal reflection phenomenon will occur on the second refractive surface. According to geometric analysis, the wedge angle should be less than 17.9${}^{{}^{\circ}}$. Let the wedge angle change from 0${}^{{}^{\circ}}$ to 17.9${}^{{}^{\circ}}$ to observe the change of the values of *y*${}_{1}$ and *y*${}_{2}$. As shown in Figure 10, the blue line is the *y*${}_{1}$ value curve, and the red line is the *y*${}_{2}$ value curve. According to the actually size of the CCD, it can be seen that the prism wedge angle should be chosen to be ~6~13.5${}^{{}^{\circ}}$.

### 2.3. Compensation for Focal Length Changes Based on Finite Point Imaging Changes on the Axis

Now, the change of the imaging point of the finite distance point on axis through the optical system is investigated. According to the position change of the image point of the object point on axis after the image separation, the distance between the CCD plane and the lens is compensated.

Samely, as the upper and lower sides of the axis are symmetrical in the optical path, only the optical path change on upper side is analyzed. The intersection of the two rays passing through the lens and the prism system is the new image point position after the image is separated.

As shown in Figure 11, point P is on axis, and the two-dimensional plane coordinate system is established with the lens center as the origin. The lens aperture is *D*${}_{1}$, the prism aperture is *D*${}_{2}$, the wedge angle is θ, and the prism-to-lens spacing is *d*. A${}_{1}$,B${}_{1}$,C${}_{1}$ is the point at which ray 1 turns through the optical path of the optical system, and A${}_{2}$,B${}_{2}$,C${}_{2}$ is the point at which ray 2 turns through the optical path of the optical system. P’ is the image point position of the P point when there is no prism, and P” is the image point position after the image is separated when the prism is added. The derivation process is similar to Equation (Equation 1). The parallel ray incident in Equation (Equation 1) is changed to the finite distance point source incidence, which can be combined with the single-lens Gaussian imaging formula to obtain *l’*, and change the incident angle into α${}_{1}$ = arctan($\frac{y_{A1}}{l^{\prime }}$) to obtain two outgoing rays (straight line C${}_{1}$P” and straight line C${}_{2}$P”) equation:(10)$$\left\{\begin{array}{c} x_{P^{\prime \prime }}=\frac{y_{C2}-y_{C1}+\tan\left(r_{2}-\theta \right)x_{C2}-\tan\left(r_{1}-\theta \right)x_{C1}}{\tan\left(r_{2}-\theta \right)-\tan\left(r_{1}-\theta \right)}\\ y_{P^{\prime \prime }}=y_{C1}-\tan\left(r_{1}-\theta \right)\left(\frac{y_{C2}-y_{C1}+\tan\left(r_{2}-\theta \right)x_{C2}-\tan\left(r_{1}-\theta \right)x_{C1}}{\tan\left(r_{2}-\theta \right)-\tan\left(r_{1}-\theta \right)}-x_{C1}\right) \end{array}\right.$$

We use matlab to simulate the optical circuit. The simulation experiment parameters are set according to the actual situation. The lens with focal length *f* = 8.5 mm is used. The single lens and prism aperture are both 12.76 mm, the prism wedge angle is 10${}^{{}^{\circ}}$. Make the point distance on the axis *l* = −20, and observe the change of the image point, as shown in Figure 12. It can be seen from the figure that after the point on axis’ image is separated by the prism system, the position of the image point is closer to the back surface of the lens than the original image point, that is, the image distance is shortened. Change the object distance and observe the change of the image point. Considering the actual application process, the distance between the measured object and the camera lens is as large as 10 cm~3 m. Substituting *l* = −100~−3000 mm, observe the change of the position of the image point. Figure 13 shows the change in image point when the lens focal length is 8.5 mm with a 10${}^{{}^{\circ}}$ prism. The blue star dots represent the position coordinates of the point on the axis that is imaged by a single lens, and the red star points indicates the position of the point on the axis after adding the prism. The distance between the image plane and lens varies from 8.5 mm to 9 mm without prism, whereas if there is a prism, it varies from 5.3 mm to 6.3 mm. Therefore, if focal length compensation is to be performed inside the optical system, there are two methods: moving the CCD plane forward, reducing the distance between the CCD surface and the rear surface of the lens so that it is just at the ideal image point position; increasing the focal length of the lens, making the image point move back to the CCD plane position. We use the zoom lens to reduce the distance between the lens and the CCD plane by moving the position of the back focal plane while the focal length of the lens is constant, compensating for the focal length change caused by the rear of the prism.

### 2.4. Imaging Clarity Evaluation

To more accurately describe and evaluate the imaging quality of new optical systems, we need to perform Image Quality Assessment(IQA) [25]. The image quality was quantitatively evaluated by using no-reference evaluation algorithms [26,27]. In this paper, Laplacian gradient function [28], Tenengrad gradient function, and gray-scale variance function evaluation algorithms are used to quantitatively compare the imaging quality of the prism postposition mirror binocular system before and after focal length compensation.

The Laplacian gradient function uses the Laplacian operator to extract the horizontal and vertical gradient values [29]. The Laplacian operator is defined as follows.
(11)$$L=\frac{1}{6}\left[\begin{array}{ccc} 1 & 4 & 1\\ 4 & -20 & 4\\ 1 & 4 & 1 \end{array}\right]$$
The definition of image sharpness based on Laplacian gradient function is as follows.
(12)$$D\left(f\right)=\sum_{y}\sum_{x}\left|G(x,y)\right|\left(G\left(x,y\right)>T\right)$$
where G(*x*, *y*) is the convolution of the Laplacian operator at the pixel point (*x*, *y*).

The Tenengrad gradient function [30] is basically the same as the Laplacian gradient function, just replacing the Laplacian operator with the Sobel operator:(13)$$g_{x}=\frac{1}{4}\left[\begin{array}{ccc} -1 & 0 & 1\\ -2 & 0 & 2\\ -1 & 0 & 1 \end{array}\right],g_{y}=\frac{1}{4}\left[\begin{array}{ccc} 1 & 2 & 1\\ 0 & 0 & 0\\ -1 & -2 & -1 \end{array}\right]$$
The gray-scale variance function [31] uses the gray-scale variation as the basis for the focus evaluation. The formula is as follows.
(14)$$D\left(f\right)=\sum_{y}\sum_{x}\left(|f(x,y)-f(x,y-1)|+|f(x,y)-f(x+1,y)|\right)$$

## 3. Experiment and Analysis

### 3.1. Construction of the Catadioptric Component Postposition System

According to the above theoretical analysis, the catadioptric structure postposition optical system is constructed. We use the composite target as the object to be tested, the optical support to fix the object(target) on the optical platform, and a computer to store image data. The processor is an Intel Core i7-7700HQ with 64-bit operating system. The single-camera mirror binocular vision system consists of a single camera, a prism and a zoom lens. The experiment uses a zoom lens with focal length of ~6–12 mm for focal length compensation imaging experiment. In the experiment, when the single lens is normally shot, the prism is directly added to take an image without focal length compensation, and then the back focal plane is adjusted to capture the image after compensating focal length. The camera, prism, and zoom lens parameters are shown in Table 4, and the experiment set-up is shown in Figure 14.

The zoom lens uses ZLKC’s megapixel HD lens, model VM06012MP, and the focal length can be changed from 6 to 12 mm by manual adjustment. The lens has three adjustment rings that adjust the focus, aperture, and back focal plane.

### 3.2. Comparative Analysis of Image Clarity before and after Prism Postposition Focal Length Compensation

In the experiment, corresponding to the parameters of the simulation experiment, we fixed the focal length into 8.5 mm without prism, and the aperture and the back focal plane are adjusted to achieve the best image clarity. Keep the three rings fixed, mount the prism on the back surface of the zoom lens, assembled into a catadioptric component postposition mirror binocular stereo vision system with the camera. Then, we use the single camera to capture a set of target images. This group of images is captured when the prism is added directly and the focus is not compensated. Next, the back focal plane is separately adjusted, the distance between the CCD plane and the lens is changed, and a set of target images are captured after compensating for the focal length change. This group is the image after the focal length compensation, and the two groups of images are shown in Figure 15. Then, we adjust the focal length ring, observe the imaging effect after the prism is placed on different focal length lenses, repeat above operations, and then take the target images before and after the focal length compensation of different focal length lenses. Figure 16 shows the mirrored binocular imaging pairs before and after the focal length compensation at focal lengths of 6.34 mm, 10.21 mm, and 11.12 mm, with a total of 9 pairs in each group. The above image was evaluated by three image clarity quantification method, and the image clarity before and after the focal length compensation was compared. The image clarity quantitative evaluation comparison value is shown in Table 5, Table 6 and Table 7. In these tables, “U” represents the clarity values of the uncompensated images and “C” means compensated images. Figure 17 is a chart of the comparison values, and Figure 18 shows the mean values evaluating the clarity of the images.

From the comparison of the image clarity before and after the above four sets of focal length compensation, it can be seen form Figure 18 that after the prism is placed behind the lens with different focal lengths, the image clarity after the focal length compensation is still not ideal, but the image quality is significantly improved than no compensation. It can be seen that by moving the back focal plane, reducing the distance between the CCD plane and the lens can compensate for the focal length change caused by the postposition of the prism and improve the imaging quality.

### 3.3. Calibration Experiment of the Catadioptric Component Postposition in Mirror Binocular

To prove the feasibility of the mirror binocular sensor with post-catadioptric component, we collected a series of dot target images and completed the calibration the experiment. The calibration accuracy determines the measurement accuracy, and the calibration accuracy is determined by the optimization method [32], whose accuracy is given as a pixel fraction. The original image of the target is shown in Figure 19. The extracted feature points images of two virtual cameras is shown in Figure 20, and the calibration results are listed in Table 8 and Table 9. In Table 9, $\left(\alpha_{x},\alpha_{y}\right)$ are the normalized focal lengths of two virtual cameras, $\left(u_{0},v_{0}\right)$ are the principal point coordinates, $\left(k_{1},k_{2},k_{3}\right)$ represent the radial distortion coefficients, and $\left(p_{1},p_{2}\right)$ represent the tangential distortion coefficients. Table 9 lists the relationship between the two virtual cameras. We can see that the back-reprojection error and forward-reprojection error of each images of the two virtual cameras are both small in Figure 21, which meets general visual measurement requirement. The mean back-reprojection errors of the left and right virtual camera are 0.012 pixel and 0.012 pixel. The mean forward-reprojection errors of the left and right virtual camera are 0.022 mm and 0.033 mm.

### 3.4. Discussion and Future Work

This article focuses on imaging analysis. We concentrate on the effect of catadioptric postposition on the focal length and imaging surface, using a zoom lens to reduce distance between the CCD and lens. In future research, we will further study the focal length compensation of the imaging system, such as the calibration of the aberrations. Also, we intend to use a special optical component whose refractive index is different at different parts to make compensation strategies in the subsequent studies. It will make focal length compensation achieve better results.

Imaging is the foundation of vision sensors. We proposed a totally novel mirror binocular system, which is significant for the development of vision sensors, and the imaging issues of the novel system need to be discussed in depth. Therefore, the detailed three-dimensional reconstruction contents are planned to be discussed in future work.

## 4. Conclusions

In this paper, a novel mirror binocular system with the catadioptric component postposition is proposed. After the prism is placed behind the lens, it is photographed by a single camera to obtain two images of the measured object in one picture. The simulation experiment of the prism postposition system was carried out. The influence of prism parameters on the focal length of the optical system was analyzed, and the method of focal length compensation was proposed. After the prism is placed, the focus will be moved forward and the focal length will be reduced. The method of reducing the distance between the CCD plane and the lens can be used to compensate. The Laplacian gradient function, Tenengrad gradient function and gray-scale variance algorithms is used to quantitatively evaluate the image clarity. It is verified by experiments that the compensated image quality is better than no compensation image, so the focal length compensation can be realized. Besides, the imaging and calibration experiment has been implemented, which demonstrated the feasibility of the designed sensor. Therefore, the catadioptric component postposition system can obtain a mirror binocular image, which is perfectly simultaneous and greatly reduces the cost and the operation as well as being more convenient than the conventional mirrored binocular system.

## Figures and Tables

**Figure 1 sensors-19-05309-f001:**
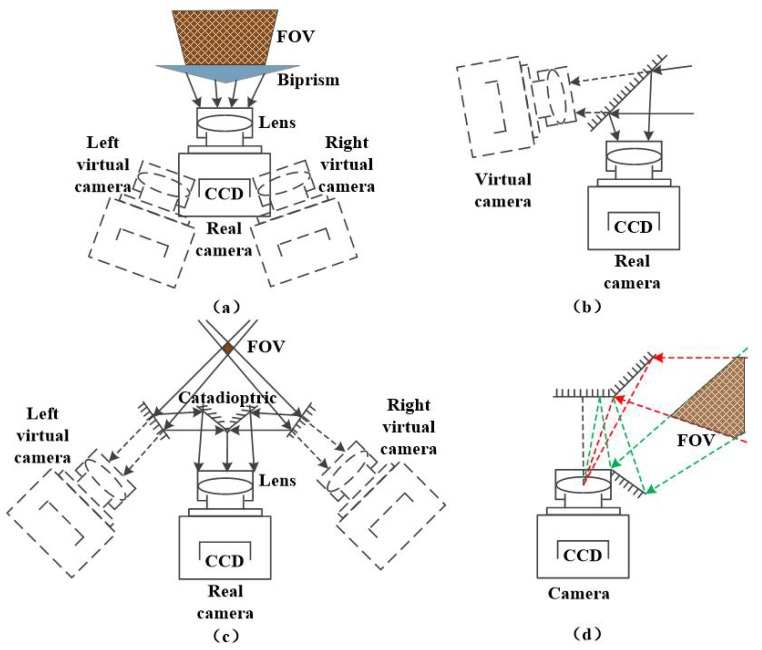
Traditional mirror binocular diagram. (**a**) Prism in front of camera lens. (**b**) Single plane mirror in front of lens. (**c**) Four plane mirror in front of lens. (**d**) Quadrangular pyramid mirror in front of lens.

**Figure 2 sensors-19-05309-f002:**
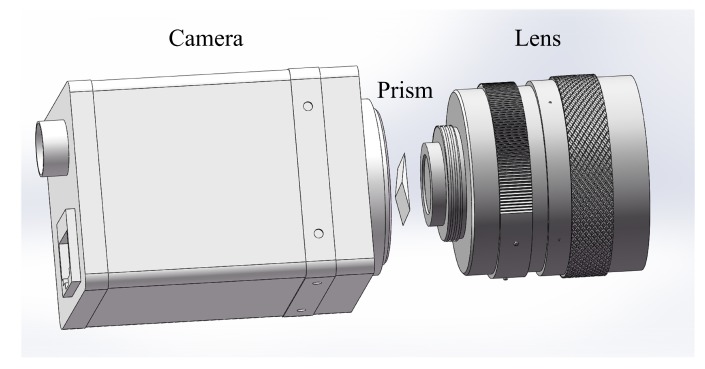
Catadioptric postposition schematic diagram.

**Figure 3 sensors-19-05309-f003:**
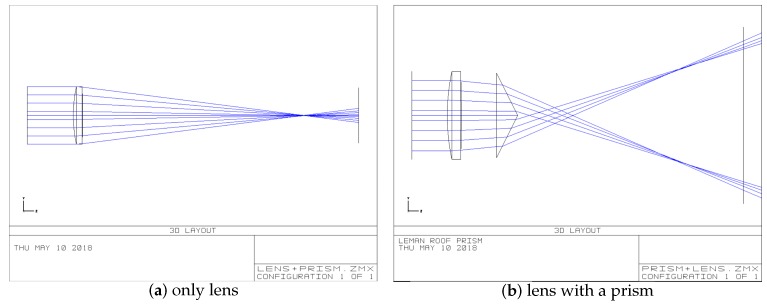
Comparison with single lens and lens with a prism.

**Figure 4 sensors-19-05309-f004:**
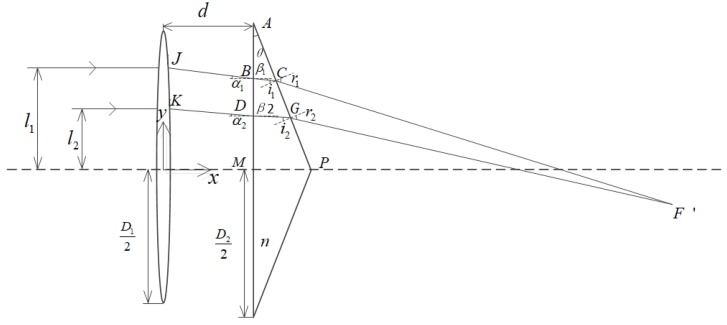
Parallel light incident derivation diagram.

**Figure 5 sensors-19-05309-f005:**
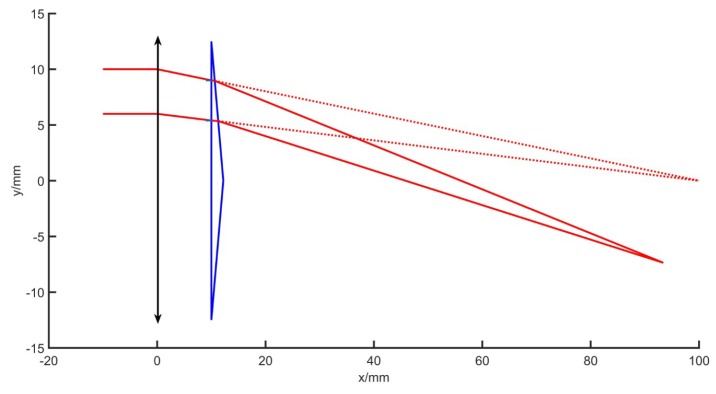
Schematic diagram of optical path change.

**Figure 6 sensors-19-05309-f006:**
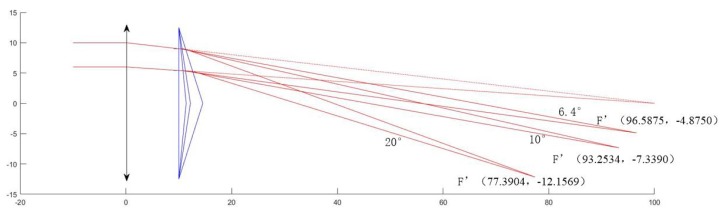
Optical path simulation in the change of prism wedge angle.

**Figure 7 sensors-19-05309-f007:**
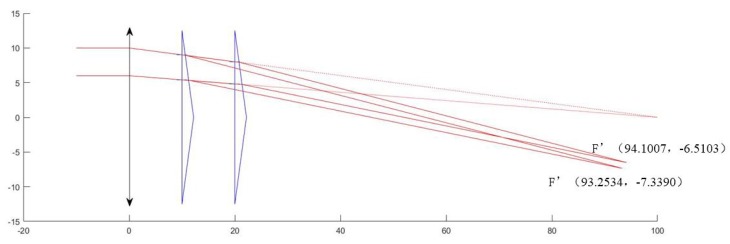
Optical path simulation in the change of the distance between prism and lens.

**Figure 8 sensors-19-05309-f008:**
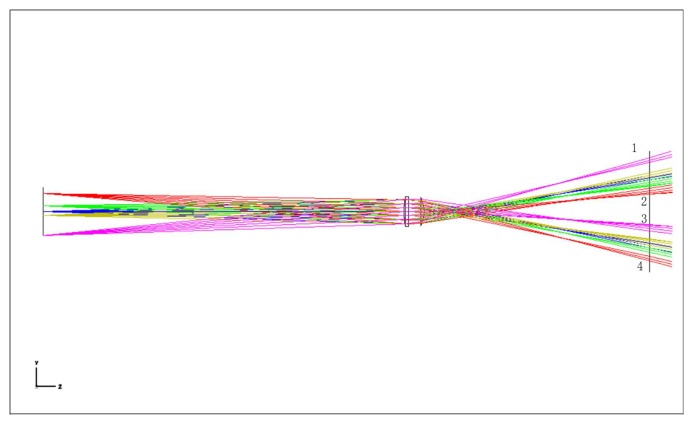
Layout 3D of point light path in finite distance.

**Figure 9 sensors-19-05309-f009:**
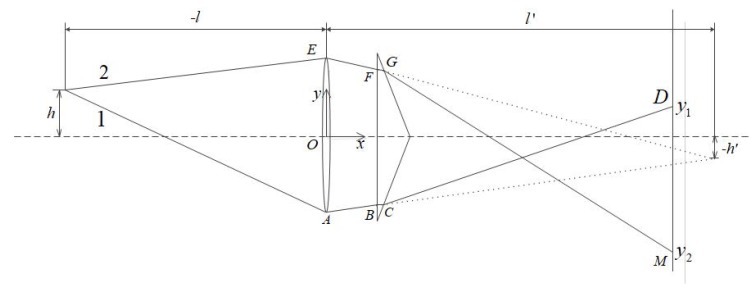
Light path diagram of point in finite distance.

**Figure 10 sensors-19-05309-f010:**
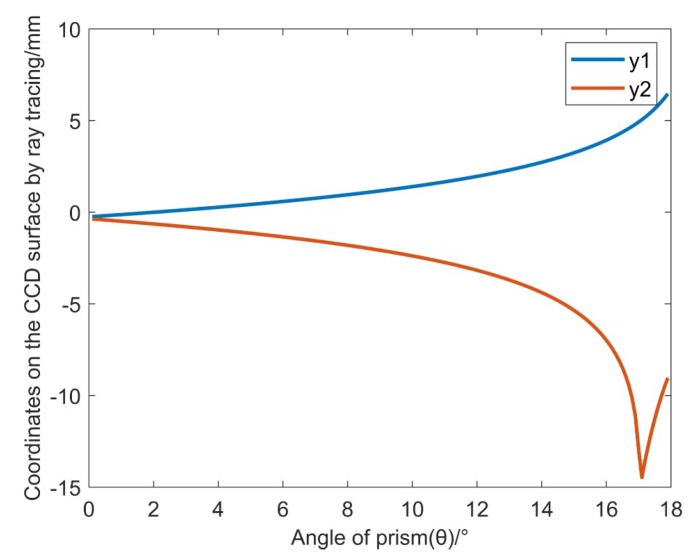
Value of *y*${}_{1}$ and *y*${}_{2}$ varies with the change of the prism wedge angle.

**Figure 11 sensors-19-05309-f011:**
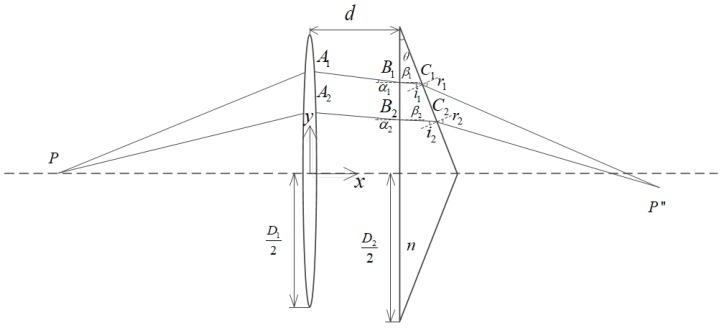
Schematic diagram of point ray tracing on the axis.

**Figure 12 sensors-19-05309-f012:**
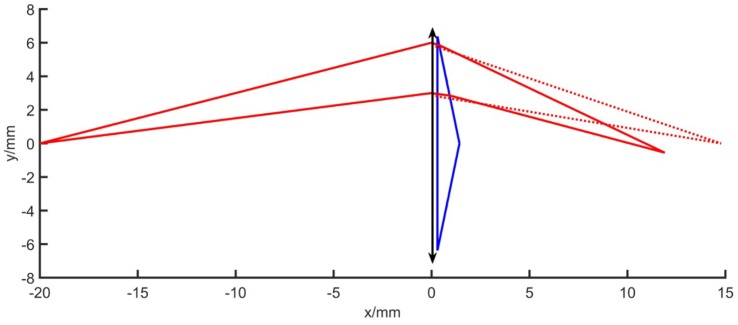
*l* = −20, θ = 10${}^{{}^{\circ}}$ optical path simulation.

**Figure 13 sensors-19-05309-f013:**
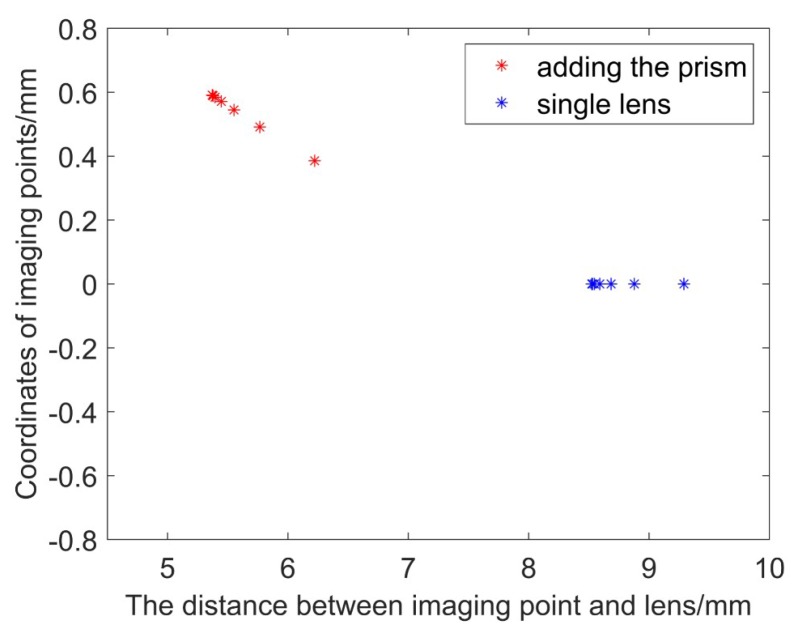
Image point position (*l* = −100~−3000, θ = 10${}^{{}^{\circ}}$, *f* = 8.5 mm).

**Figure 14 sensors-19-05309-f014:**
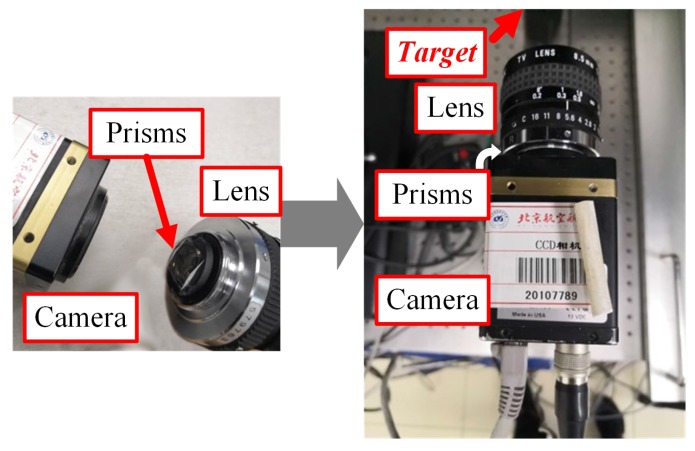
The experiment set-up.

**Figure 15 sensors-19-05309-f015:**
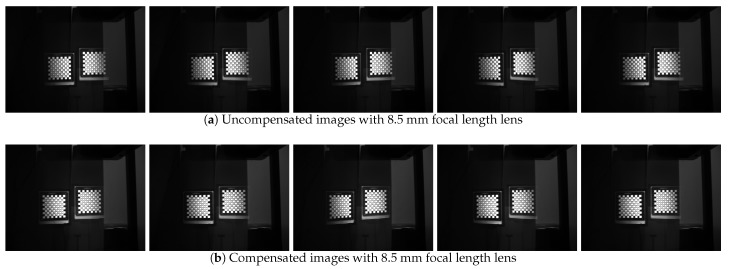
Comparison between uncompansated and compansated images with 8.5 mm focal length lens.

**Figure 16 sensors-19-05309-f016:**
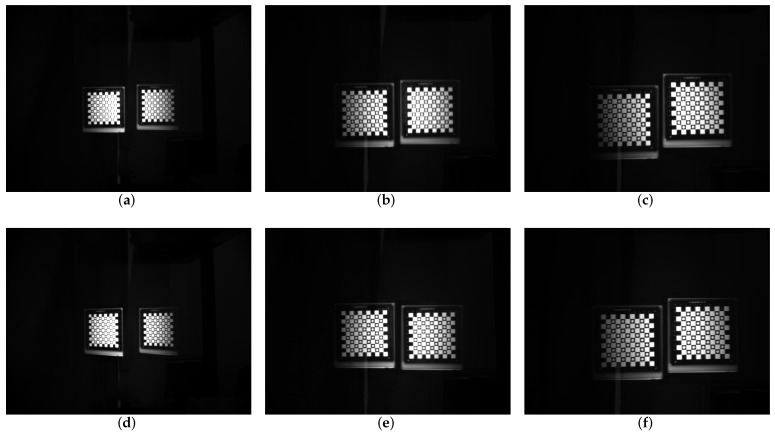
Images before and after compensation comparision. (**a**) Uncompensated image with *f*${}_{1}$ focal length lens. (**b**) Uncompensated image with *f*${}_{2}$ focal length lens. (**c**) Uncompensated image with *f*${}_{3}$ focal length lens. (**d**) Compensated image with *f*${}_{1}$ focal length lens. (**e**) Compensated image with *f*${}_{2}$ focal length lens. (**f**) Compensated image with *f*${}_{3}$ focal length lens.

**Figure 17 sensors-19-05309-f017:**
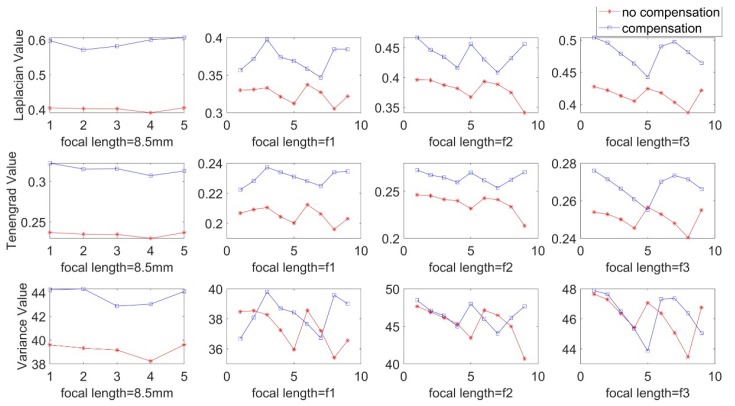
Different focal length uncompensated and compensated Articulation Value comparison.

**Figure 18 sensors-19-05309-f018:**
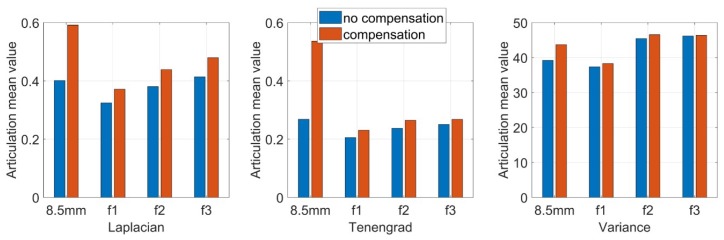
Different focal length uncompensated and compensated Articulation mean value comparison.

**Figure 19 sensors-19-05309-f019:**
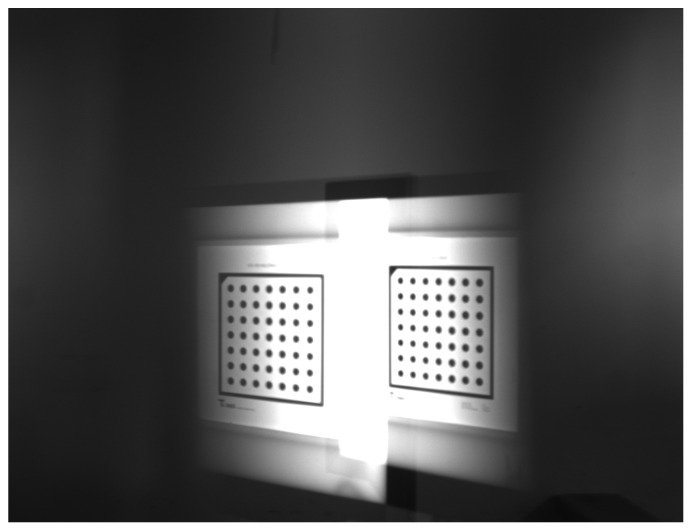
Dot target image.

**Figure 20 sensors-19-05309-f020:**
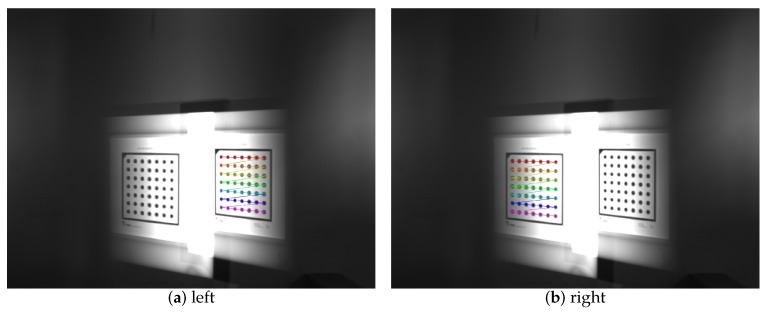
Feature points of in left and right images.

**Figure 21 sensors-19-05309-f021:**
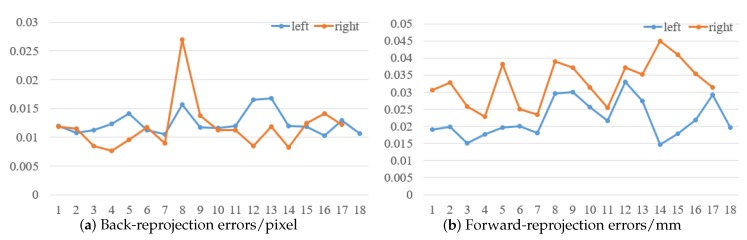
Reprojection errors.

**Table 1 sensors-19-05309-t001:** New lens–prism group parameters setting.

Number	Type	Radius	Thickness	Material	Aperture
0	−∞	∞	−	−
1	Standard	∞	10.000	−	12.500
2	Standard	60.171	4.000	Glass(BK7)	12.500
3	Standard	−368.450	120.000	−	12.321
4	Non-sequence model	∞	−	Glass(BK7)	12.321
5	Standard	∞	0	−	12.500

**Table 2 sensors-19-05309-t002:** Changes of optical system parameters.

D${}_{1}$/mm	D${}_{2}$/mm	*f*/mm	*d*/mm	θ ${/}^{{}^{\circ}}$	F’/mm
25	25	100	10	20	(77.3904,−12.1569)
25	25	100	10	10	(93.2534,−7.3390)
25	25	100	10	6.4	(96.5875,−4.8750)
20	25	100	10	10	(93.2534,−7.3390)
25	25	100	20	10	(94.1007,−6.5103)

**Table 3 sensors-19-05309-t003:** New lens–prism group parameters setting with finite point incidence.

Number	Type	Radius	Thickness	Material	Aperture
0	−∞	300	−	−
1	Standard	60.171	4.000	Glass(BK7)	12.500
2	Standard	−368.450	120.000	−	12.321
3	Non-sequence model	∞	−	Glass(BK7)	12.321
4	Standard	∞	0	−	12.500

**Table 4 sensors-19-05309-t004:** Single-camera mirror binocular vision system set-up.

Key Component	Configuration
Camera	Type: IMPERX IGV-B1610M-SC000 Sensors: CCD,1/1.8” Resolution: 1628 × 1236 Pixel size: 4.4 μm × 4.4 μm
Zoom lens	Type: ZLKC VM06012MP Operation: Manual Aperture range: F1.6-C Focal length: 6 mm~12 mm
Prism	Wedge Angle: 10${}^{{}^{\circ}}$ Size: 12.76 mm × 12.76 mm × 1.12 mm

**Table 5 sensors-19-05309-t005:** Images clarity, Laplacian evaluation method.

Number	8.5 mm		f${}_{1}$		f${}_{2}$		f${}_{3}$	
	U	C	U	C	U	C	U	C
1	0.404916	0.597787	0.329958	0.356612	0.396266	0.466913	0.427831	0.503828
2	0.402704	0.571568	0.33086	0.371547	0.395573	0.446222	0.422204	0.495848
3	0.402364	0.582054	0.33305	0.397381	0.38742	0.434543	0.413441	0.478734
4	0.390978	0.600274	0.321311	0.373777	0.381786	0.416509	0.405406	0.463751
5	0.405271	0.60677	0.312168	0.368947	0.367252	0.456152	0.424629	0.442641
6	−	−	0.337233	0.358563	0.393361	0.430181	0.418071	0.490165
7	−	−	0.327276	0.346849	0.388683	0.408106	0.403412	0.497333
8	−	−	0.305368	0.384678	0.37485	0.432449	0.387374	0.481165
9	−	−	0.321917	0.384528	0.341102	0.456143	0.422035	0.464623
mean	0.401249	0.591691	0.324349	0.371431	0.380699	0.438580	0.413822	0.479788

**Table 6 sensors-19-05309-t006:** Images clarity, Tenengrad evaluation method.

Number	8.5 mm		f${}_{1}$		f${}_{2}$		f${}_{3}$	
	U	C	U	C	U	C	U	C
1	0.237058	0.322626	0.206773	0.222388	0.246308	0.272646	0.253961	0.276075
2	0.234988	0.315484	0.209144	0.228171	0.245331	0.267377	0.252828	0.271486
3	0.234645	0.315858	0.21057	0.237282	0.241437	0.264822	0.250102	0.266422
4	0.229827	0.30731	0.204355	0.233996	0.239944	0.259673	0.245449	0.260839
5	0.236956	0.313009	0.200203	0.231025	0.23165	0.270032	0.256534	0.255067
6	−	−	0.212365	0.228074	0.242763	0.261756	0.252835	0.270176
7	−	−	0.206249	0.224787	0.24121	0.253707	0.247988	0.273451
8	−	−	0.196007	0.234024	0.23366	0.262363	0.240395	0.27147
9	−	−	0.203061	0.234544	0.213349	0.270594	0.255026	0.266252
mean	0.234695	0.314857	0.205414	0.230477	0.237295	0.264774	0.250569	0.267972

**Table 7 sensors-19-05309-t007:** Images clarity, Variance evaluation method.

Number	8.5 mm		f${}_{1}$		f${}_{2}$		f${}_{3}$	
	U	C	U	C	U	C	U	C
1	39.5996	44.2464	38.4736	36.6813	47.6726	48.4819	47.6474	47.8818
2	39.3334	44.3136	38.5472	38.0927	46.9064	47.0809	47.2903	47.6474
3	39.1678	42.8679	38.2744	39.7999	46.1471	46.4467	45.4121	45.3387
4	38.2404	43.027	37.2483	38.6948	45.3321	48.0055	47.0678	43.8563
5	39.6101	44.094	35.9539	38.368947	43.4656	48.0055	47.0678	43.8563
6	−	−	38.5745	37.6588	47.1563	45.9814	46.3777	47.3133
7	−	−	37.2112	36.7383	46.4811	44.0478	45.0714	47.3574
8	−	−	35.415	39.5831	45.017	46.1331	46.7654	45.0493
9	−	−	36.5621	39.0101	40.6953	47.6721	46.7654	45.0493
mean	39.1903	43.7092	37.3622	38.2982	45.4304	46.5377	46.1617	46.3686

**Table 8 sensors-19-05309-t008:** Calibration results of single virtual cameras.

Parameters	$\left(\alpha_{x},\alpha_{y}\right)$/(pixel)	$\left(u_{0},v_{0}\right)$/(pixel)	$\left(k_{1},k_{2},k_{3}\right)$/(1/pixel${}^{2}$)	$\left(p_{1},p_{2}\right)$/(1/pixel${}^{2}$)
Left	(1938.31,1938.67)	(898.21,587.39)	(−0.083,−3.57,2.48)	(0.0033,0.0021)
Right	(1945.71,1935.00)	(1083.07,611.65)	(−0.84,15.03,−1.81)	(0.028,0.0065)

**Table 9 sensors-19-05309-t009:** Calibration results of structure parameters of two virtual cameras.

R	T
$\left(\begin{array}{ccc} 0.0034 & 0.4626 & -0.8865\\ 0.2169 & 0.8658 & 0.4510\\ 0.9762 & -0.1908 & -0.1033 \end{array}\right)$	$\left(\begin{array}{c} 228.40\\ -178.52\\ 331.88 \end{array}\right)$
**E**	**F**
$\left(\begin{array}{ccc} -246.27 & -253.27 & -131.23\\ -224.08 & 197.12 & -270.63\\ 48.94 & 280.33 & -55.26 \end{array}\right)$	$\left(\begin{array}{ccc} 5.246\times 10^{7} & 5.3947\times 10^{7} & -2.462\times 10^{4}\\ 4.7998\times 10^{7} & -4.2216\times 10^{7} & 9.4046\times 10^{7}\\ -1.0646\times 10^{3} & -1.4877\times 10^{3} & 1.0000 \end{array}\right)$

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
