# Peer review of "Effect of Catadioptric Component Postposition on Lens Focal Length and Imaging Surface in a Mirror Binocular System"

_sensors, 2019, doi:10.3390/s19235309_

Round 1
Reviewer 1 Report
This paper introduces a catadioptric postposition system, which places the prism behind the lens to achieve mirror binocular imaging. The design of this imaging system is interesting, but there are some concerns that need to be addressed.
1. The meaning of the UAV representative at the beginning of the article should be clearly stated.
2. There is a grammatical error in the last sentence of abstract.
3. Many sentences in the article are very long and complex, which are difficult to be understood clearly and simply by readers, such as lines 26-29.
4. Many of the sentences in the article are not rigorous enough. It is suggested that the author should thoroughly check the sentences and grammar of the article.
5. What is the meaning of the ‘it’ in line 77?
6. ’on’ in line 79 should be replaced by ‘of’.
7. The description of lines 79-80 has grammatical problems.
8. ’which’ in line 79 should be replaced by ‘in which’.
9. When the author introduces a prism to obtain binocular vision, it also affects the original imaging system. Does the new system bring loss in imaging systems such as aberrations (such as coma, spherical aberration)? The author can simulate the aberration data of the new system in the simulation to evaluate the imaging quality of the system.
10. Although the author uses a zoom lens to compensate for the focal length, the focal lengths that need to be compensated for different parts of the same picture are different. How do the authors make compensation strategies?
11. Only schematic setup of the system is shown, the experiment setup should be included in section 3 for a better understanding of the system integration.
12. Since the system is specifically design for mirror binocular, it’s mandatory to evaluate the three-dimensional reconstruction result to further investigate the feasibility of the presented design.
Reviewer 2 Report
Manuscript number: sensors-641947
Title: Effect of catadioptric component postposition on lens focal length and imaging surface in mirror binocular
Authors: Fuqiang Zhou, Yuanze Chen, Mingxuan Zhou and Xiaosong Li
The catadioptric postposition system is an interesting task. I agree with the concepts explained in the paper. However, the paper lacks some matters to establish the powerful of the proposed method. In the paper, the fundamentals and ray-tracing are well described. But, other sections should be improved. Therefore, comments should be included in the manuscript.
1.- The main aim of the proposed technique is to obtain a binocular system via prism. In a binocular system, the first camera captures the occluded regions of the second camera and the second camera captures the occluded regions of the first camera [1]. This statement is not described in the paper. Comments about these matters should be included.
2.- The calibration accuracy determines the measurement accuracy. And the calibration accuracy is determined by the optimization method [2], whose accuracy is given as a pixel fraction. However, the manuscript does not describe the optimization procedure. Comments about these matters should be included.
3.- The main parameter in a vision system is the image distortion coefficients. These coefficients are not described in the paper. Comments about these matters should be included.
4.- The proposed method claims that low cost. However, the cost of the prism is not low. What is the real price difference between a camera and a prism. Comments about these matters should be included.
5.- The most important procedure to determine the powerful of the system is to determine three-dimensional coordinates. Comments about these matters should be included.
[1] JOMO, Vol.63, p.1219-1232, (2016).
[2] OLEN, Vol. 105, p. 75–85, (2018).
